# Comprehensive Analysis and Drug Modulation of Human Endogenous Retrovirus in Hepatocellular Carcinomas

**DOI:** 10.3390/cancers15143664

**Published:** 2023-07-18

**Authors:** Ya-Sian Chang, Ming-Hon Hsu, Chin-Chun Chung, Hong-Da Chen, Siang-Jyun Tu, Ya-Ting Lee, Ju-Chen Yen, Ta-Chih Liu, Jan-Gowth Chang

**Affiliations:** 1Center for Precision Medicine, China Medical University Hospital, Taichung 40447, Taiwan; t25074@mail.cmuh.org.tw (Y.-S.C.); t35492@mail.cmuh.org.tw (M.-H.H.); t35121@mail.cmuh.org.tw (C.-C.C.); t35882@mail.cmuh.org.tw (H.-D.C.); t34752@mail.cmuh.org.tw (S.-J.T.); t23701@mail.cmuh.org.tw (Y.-T.L.); t24399@mail.cmuh.org.tw (J.-C.Y.); 2Epigenome Research Center, China Medical University Hospital, Taichung 40447, Taiwan; 3Department of Laboratory Medicine, China Medical University Hospital, Taichung 40447, Taiwan; 4School of Medicine, China Medical University, Taichung 40402, Taiwan; 5Department of Hematology-Oncology, Chang Bing Show Chwan Memorial Hospital, Changhua 50544, Taiwan

**Keywords:** Taiwanese hepatocellular carcinoma, Telescope, host genes, total RNA sequencing, human endogenous retrovirus

## Abstract

**Simple Summary:**

Human endogenous retrovirus (HERV) plays important roles in the development of cancer, and most studies use data from The Cancer Genome Atlas to analyze the whole HERV alterations in cancer cells. For HCCs, most studies have focused on LINE-1 and specific HERVs to explore their importance. In this study, we used our total RNA sequencing data from 254 Taiwanese HCCs and many bioinformatic tools to analyze HERV alterations, and then explored the correlations between HERV activation and pathways and certain gene panels. Unique pathways for a higher expression of survival-related HERVs including immunity and infection, lipid and atherosclerosis, MAPK and NF-kB signaling, and cytokine-cytokine receptor interaction pathways were activated; the mRNA surveillance pathway, nucleocytoplasmic transport, ribosome biogenesis, and transcriptional misregulation in cancer pathways were suppressed. We found that many overexpressed HERV-related nearby genes were correlated with high HERV activation and poor survival. The implementation of comprehensive and integrated approaches to assess HERV expression and their association with specific pathways is poised to offer novel companion diagnostics and therapeutic strategies for HCC.

**Abstract:**

Background: Human endogenous retroviruses (HERVs) play an important role in the development of cancer and many diseases. Here, we comprehensively explored the impact of HERVs on hepatocellular carcinomas (HCCs). Methods: We employed Telescope to identify HERVs and quantify their expression in the total RNA sequencing data obtained from 254 HCC samples, comprising 254 tumor tissues and 34 matched normal tissues. Results: In total, 3357 locus-specific activations of HERVs were differentially expressed, and 180 were correlated with patient survival. Using these 180 HERVs for classification, we found four subgroups with survival correlation. Higher expression levels of the 180 HERVs were correlated with poorer survival, while age, AFP, some mutations, and copy and structural variants differed among subgroups. The differential expression of host genes in high expression of these 180 HERVs primarily involved the activation of pathways related to immunity and infection, lipid and atherosclerosis, MAPK and NF-kB signaling, and cytokine–cytokine receptor interactions. Conversely, there was a suppression of pathways associated with RNA processing, including nucleocytoplasmic transport, surveillance and ribosome biogenesis, and transcriptional misregulation in cancer pathways. Almost all genes involved in HERV activation restriction, KRAB zinc finger proteins, RNA nucleocytoplasmic transport, stemness, HLA and antigen processing and presentation, and immune checkpoints were overexpressed in cancerous tissues, and many over-expressed HERV-related nearby genes were correlated with high HERV activation and poor survival. Twenty-three immune and stromal cells showed higher expression in non-cancerous than cancerous tissues, and seven were correlated with HERV activation. Small-molecule modulation of alternative splicing (AS) altered the expression of survival-related HERVs and their activation-related genes, as well as nearby genes. Conclusion: Comprehensive and integrated approaches for evaluating HERV expression and their correlation with specific pathways have the potential to provide new companion diagnostics and therapeutic strategies for HCC.

## 1. Introduction

Hepatocellular carcinoma (HCC) is one of the most common and deadly cancers worldwide [1]. Many factors are involved in the development of HCC, such as chronic viral infection, alcohol abuse, diabetes mellitus (DM), obesity, metabolic diseases, hemochromatosis, and genetic factors [2,3]. Non-infectious HCC is increasing in developed countries due to increased obesity rates, DM, and metabolic diseases [4]. These risk factors induce liver injury, resulting in progressive inflammation and making liver cells enter a cycle of cell death and regeneration, followed by the development of somatic mutation and chromosomal instability [5,6]. Surgery is the major approach for the resected HCC; however, many HCCs found at the unresected stage require chemotherapy or targeted therapy. Despite the many potential therapeutic targets, few drugs have shown clinically promising effects, and most of these drugs increased survival by only a few months, indicating the need for new targets for HCC treatment [3,6].

Endogenous retroviruses (ERVs) are retrotransposons, a type of transposable element (TE) that spreads throughout the genome via a copy-and-paste mechanism. Human endogenous retroviruses (HERVs) are residents in the human genome and are located on ~8% of genomic DNA. There are about 100 HERV families based on common features in the human genome [7,8]. Reactivation of HERVs is found in many cancers and can influence tumor genome stability [9,10,11,12]. HERVs can serve as alternative promoters or enhancers for nearby genes in malignant cells, inducing both tumor suppressor gene (TSG) downregulation and oncogene upregulation, and cryptic transcription start sites within HERVs can be employed to produce aberrant protein-coding mRNAs [13,14,15]. These alterations result in cancer development, progression, metastasis, immune alterations, and chemoresistance [9,10,11,12,14,15,16,17,18]. Retrotransposons are correlated with the development of HCC, but most studies have focused on LINE-1, and rarely on whole HERVs [19,20].

Reactivation of HERVs in cancer cells may result in a viral mimicry state, the generation of highly tumor-specific antigens and expression of long terminal repeat (LTR)-activated transcripts, and subsequent implications for cancer immunotherapy [21]. The reactivation of HERVs using demethylation drugs to induce neoantigens and antiviral-like immunity has become a new approach for cancer treatment [22,23,24,25,26]. For example, DNA-demethylation agents have shown clinical anti-tumor efficiency by inducing transcription of endogenous dsRNAs that activate the viral recognition and interferon response pathway in colorectal cancer-initiating cells [26]. A comprehensive analysis of interactions between HCCs and HERVs is lacking. In this study, we employed Telescope to conduct a locus-specific characterization of survival-related differentially expressed (DE) HERVs in HCCs. These survival-related DE HERVs were used for molecular classification. Subsequently, we investigated the correlation between HERV subgroups and the expression levels of genes associated with HERV restriction, viral immunity, RNA transport, stemness, nearby genes, and the tumor micro-environment. Finally, we analyzed the impact of splicing-modulating drugs on the expression of HERVs.

## 2. Materials and Methods

### 2.1. Liver Samples and Clinical Data

Hepatocellular carcinoma (HCC) was determined by pathological diagnosis. Tumor and adjacent non-tumor liver tissue samples were collected and frozen at −80 °C after surgical resection at the tissue bank of China Medical University Hospital (CMUH). The tissue bank was established in 2005 and has been accredited by the Taiwanese government since 25 October 2012, making it the first hospital-based accredited tissue bank in Taiwan. This bank has collected more than 20,000 cancer tissues, including more than 20 types of cancers. The clinical data were collected from the data warehouse of CMUH; the warehouse collects clinical data, including history, laboratory, pathological, and image data from electronic medical records of more than 2 million patients over more than 20 years in CMUH. The survival time was collected from the operation date; we only selected patients who underwent an operation (254 cases) and excluded liver transplantation patients (36 cases) for survival evaluation (218 cases). Of these patients, thirty-four had matched tumor and normal tissues. This study was approved by the Ethics Committee of CMUH (CMUH 109-REC3-055), and written informed consent was obtained from all participants by the standard procedure of the CMUH tissue bank.

### 2.2. RNA Extraction and RNA Sequencing (RNA-Seq)

Total RNA was extracted from tissue samples using the NucleoSpin^®^ RNA Kit (Macherey–Nagel, Duren, Germany), following the manufacturer’s instructions. The quality, quantity, and integrity of the total RNA were evaluated using the NanoDrop 1000 spectrophotometer and Bioanalyzer 2100 (Agilent Technologies, Santa Clara, CA, USA). RNA-seq was performed as described previously [27]. Briefly, samples with an RNA integrity number > 6.0 were used for RNA-seq. A barcoded library was generated using a Total RNA Library Preparation Kit (Illumina, San Diego, CA, USA). The libraries were sequenced on a NovaSeq 6000 instrument (Illumina) using 2 × 151-bp paired-end sequencing flow cells following the manufacturer’s instructions.

### 2.3. Metatranscriptome Analysis

To characterize the microbiome composition in tumors, we used Kraken2 (v2.1.1) to analyze metatranscriptomic data; the July 2020 Kraken2 database was used to annotate human, viral, archaeal, bacterial, and fungal genes, with the National Center for Biotechnology Information (NCBI) RefSeq used as the reference sequence database, as described previously [27].

### 2.4. Retrotranscriptome and Transcriptome Quantification

After RNA-seq, the raw data were processed by the DRAGEN pipeline, and bam files were used for HERV analysis. For HERV annotation, we used the GTF annotation file from the Appendix A of Telescope [28]; the annotation is based on Ensembl hg38 release 99. We used raw counts of retrovirus transcripts from the bam files for quantification using Telescope. The transcripts per million (TPM) was utilized as the normalization method for RNA-seq data. For the analysis of differential expression (DE) between tumor and non-tumor tissues, DESeq2 (v1.36.0) [29] was used after normalization and quantification of RNA transcripts, including coding genes, non-coding genes, and HERVs in TPM. The absolute log2-fold change > 1.5 and adjusted *p* value < 0.05 were defined as differentially expressed genes. For survival-related DE analysis of retrotranscriptome and transcriptome data, we compared high (top 25%) and low (bottom 25%) expression of each HERV in tumor samples to plot Kaplan–Meier (KM) survival curves using the survminer package (v0.4.3) in R and the KM online log-rank test calculator. We used variance stabilizing transformation before principal component analysis (PCA) and hierarchical clustering, and then used the R packages ggplot (2 v3.2.2) to visualize PCA [30] and EnhancedVolcano v11.6.0 to create the volcano plot (https://github.com/kevinblighe/EnhancedVolcano, accessed on 24 August 2022) [31]. We used the circlize v0.4.10 [32] and GenomicRanges (v1.40.0) packages to construct circos plots of differentially expressed (DE) genes and HERV expression patterns. We used the Morpheus tool (http://software.broadinstitute.org/morpheus, accessed on 12 September 2022) to construct a heatmap to visualize sample-wise expression and perform hierarchical clustering for survival-related DE HERVs. We used clusterprofiler (v4.4.4) to perform gene set enrichment analysis (GSEA; v4.0.3) and pathway analysis of survival-related DE HERVs and genes. We ranked HERVs and genes using the log2FoldChange and Padj functions in DEseq2 (v1.36.0), visualized by EnhancedVolcano (v11.6.0). We used the Molecular Signatures Database (MSigDB) to identify genes and gene ontology sets [33].

### 2.5. Structure Analysis of Survival-Related HERVs, Their Genomic Regions, and Nearby Related Genes

We used the Telescope_MetaAnnotations tool to annotate and analyze the nearby genes of survival-related HERVs, as well as the exonic, intronic, promoter, and enhancer elements of coding and non-coding genes. Then, we compared gene expression between HCC and non-cancer tissues using DESeq2 (v1.30.1).

### 2.6. Molecular Classification

Molecular classification was performed using a transcriptomics-based analysis that integrated patient survival-related HERV expression panels, coupled with survival analysis, and then validated the data using The Cancer Genome Atlas (TCGA) database [34]. For molecular classification, an unsupervised approach was used. Each signature HERV was significantly up- or downregulated in one subclass relative to the other subclasses. The TPM value was used as the distance metric (one minus spearman rank correlation), and the average linkage method was applied. Signature HERVs and subjects within each class were hierarchically clustered. Significant demographic variables, clinical annotations, *TERT* promoter, *TP53* and *CTNNB1* mutations, and molecular subclasses, as well as hepatitis virus, HBV, and hepatitis C virus (HCV) infection status, are shown at the top of the heatmap.

### 2.7. Correlation Analysis of HERV Subgroups and the Expression Levels of Genes Related to HERV Restriction, Viral Immunity, RNA Transport, and Stemness

We selected gene panels from MSigDB mainly [33], and recent studies. We explored correlations among 96 HERV restriction genes, including DNA methylation, histone methylation, N6-methyladenosine (m6A) editing, A to I editing, and piwi-interacting RNA (piRNA) formation related-genes [35,36,37,38,39,40,41,42,43,44,45,46,47,48,49,50,51,52,53,54], 101 KRAP-ZFP (KZFP) genes [55,56,57], 70 nucleocytoplasmic transporting genes for RNA transport [58,59,60], 263 immune-related genes (including double stranded RNA sensor-related genes, type I and III interferon-related genes, and interferon stimulated genes) [61,62,63], 15 stimulated 3 prime antisense retroviral coding sequences (SPARCS) genes [23], 45 human leukocyte antigen (HLA)-related genes [64], 22 immune checkpoint genes [65], 54 inflammasome and inflammatory-related genes [66], 126 stemness-related genes [67,68], and 51 metabolic genes [69]. The names and functions of the genes are listed in Appendix A.

### 2.8. Tumor Microenvironment Analysis

For cell enrichment analysis of various immune and stromal cells in tumors, xCell was used. For cell enrichment analysis of various immune and stromal cells in tumors, xCell was used. Immune scores were calculated from the TPM expression matrix, as described previously [27].

### 2.9. Drug Treatment of HCC Cell Line

Data were obtained from a previous study [70]. Briefly, Huh-7 cell lines were obtained from the Bioresource Collection and Research Center, Taiwan. The cells were maintained in DMEM (solid tumor-cell lines) supplemented with 10% FBS and antibiotics (100 U/mL penicillin and 100 ug/mL streptomycin) at 37 °C in a humidified atmosphere of 5% CO_2_. In order to characterize the potential of amiloride derivative 3,5-diamino-6-chloro-*N*-(*N*-(2,6-dichlorobenzoyl)carbamimidoyl)pyrazine-2-carboxide (BS008) for HERV-based cancer therapy, we treated the Huh-7 cells (HCC-related cell line) with BS008.

### 2.10. Statistical Analyses

The clinical and pathological data were analyzed for the presence of transcriptomic alterations and HERV molecular classification using Fisher’s exact test. Wilcoxon tests were used to compare two groups of continuous data. Kaplan–Meier survival curves were visualized using survival v. 3.2.3 and surviminer v.0.4.9. Independent prognostic factors were analyzed by the Cox proportional hazards regression model. Variables in the model included gender, age, and stage. Patients with transplantation were excluded from the survival analysis.

## 3. Results

### 3.1. Clinical Data and Metatranscriptomic Analysis

The demographic data of the 254 HCC patients are shown in Appendix A, and the metatranscriptome results revealed 109 viral RNAs in 254 HCCs, including 102 HBVs and 7 HCVs (Appendix A).

### 3.2. Retrotranscriptome and Transcriptome Quantification Analyses

HERV expression differences were considered significant if the adjusted *p*-value was <0.05 and the absolute log2-fold change was >1.5. Ultimately, 3357 of 14,164 HERVs, including 1865 ERVLs, 188 ERVKs, and 1304 ERV1s, were differentially expressed (DE) between 254 tumorous and 34 nearby non-cancerous liver tissues; 3161 HERVs were upregulated, and 196 were downregulated in tumor tissues (Figure 1A; Appendix A). Additionally, there are 1752 DE HERVs in 34 paired HCC samples. Among these 1752 HERVs, 1607 HERVs (~91.7%) are shared with the 3357 HERVs found in 288 HCC samples (254 tumor samples and 34 non-tumor samples). There is good overlap between the differences in the pairs and normal versus tumor samples. The PCA of the 500 most variable HERVs showed separation of tumor and normal HERV expression data in most samples with a group of 34 non-tumor tissue samples clustered separately (Figure 1B). Moreover, we found that protein-coding genes also had a similar phenomenon (details can be provided on request).

We analyzed the correlation with survival time between high and low expression of the 3357 DE HERVs in tumor tissues and found that expression levels of 180 HERVs were significantly correlated with patient survival (Appendix A). Among the 180 survival-related DE HERVs, 36 have retrovirus-like structures (including gag, env, and pol), 54 have two retroviral components, 68 have only one retroviral component, and 22 have LTR only. The 180 survival-related DE HERVs include 113 ERVLs, 61 ERV1s, and 6 ERVKs (Figure 1C), and the chromosomal locations of these survival-related DE HERVs are shown in the middle panel under the chromosome marker in Figure 1D.

We used the HERV meta-annotations provided by Telescope to examine the structural and functional properties of DE elements and nearby or intersected genes. Of the 180 survival-related DE HERVs, 46, 116, and 18 were in intronic, intergenic and exonic regions, respectively; 12 of these 180 HERVs contained protein coding transcripts. Two HERVs in intronic regions, and one in an intergenic region, were enhancers (Appendix A). Figure 2 presents a representative example of a HERV.

We further analyzed 180 survival-related DE HERVs. The overexpression of 119 HERVs was correlated with poor survival, suggesting that these HERVs had oncogenic characteristics. Down-expression of 61 of these 180 HERVs correlated with poorer survival, suggesting a TSG-like function (Appendix A). The 81 and 18 HERVs that were up- and downregulated in cancer tissue, respectively, are summarized in Appendix A.

### 3.3. Analysis of Nearby Genes of the 180 Survival-Related DE HERVs by Telescope

We identified 262 nearby genes of the 180 survival-related DE HERVs. The expression levels of 70 nearby genes were higher in cancerous than non-cancerous tissues, including 7 genes with almost no expression in non-cancerous tissues. Sixteen nearby genes showed higher expression in non-cancerous tissues, including seven with almost no expression in tumor (*HCN1*, *SLC22A2, FAM3D*, *LRFN5, LINC01612*, *LINC02512*, and *CNBD1*) (Appendix A). We further analyzed the correlations of 262 nearby genes and identified 24 that were correlated with patient survival (Appendix A).

Six of the eighteen HERVs with higher expression in non-tumor than tumor tissues had over a 5-fold higher expression, including ERV3-16A3_I-int_1738, ERV3-16A3_I-int_1960, HERVL-int_1358, ERVL-E-int_0023, ERVL-E-int_1224, and MER50-int_0059. We also identified nearby genes with important oncogenic (*DTX4*) or TSG (*LINC01612*) functions (Appendix A).

HERV activation may result in the coactivation or disruption of nearby genes. Therefore, we analyzed the expression levels of HERVs and nearby genes. Forty-eight nearby genes were found to have higher expression in tumor tissues when 132 HERVs upregulated in cancer tissues were compared to non-tumor tissues. However, four nearby genes had a lower expression in tumor tissues under HERV activation (Appendix A).

### 3.4. Molecular Classification of 254 Taiwanese HCCs Based on the Expression Profiles of 180 Survival-Related DE HERVs

We applied the 180 HERVs to classify 254 HCCs into subgroups using an unsupervised approach. The heatmap shows normalized expression levels of the HERVs (rows) across subjects (columns) classified into four subsets, HERV-H1 (high survival-related HERV activation group 1), HERV-H2 (high survival-related HERV activation group 2), HERV-H3 (high survival-related HERV activation group 3), and HERV-L (low survival-related HERV activation) (Figure 3A). These groups were also correlated with HCC patient survival (*p* = 0.0002) (Figure 3B). After adjusting for clinical factors, the classification still demonstrated significant survival implications. HERV-L had better survival than HERV-H1 (*p* = 0.00006), HERV-H2 (*p* = 0.0254) and HERV-H3 (*p* = 0.0209), HERV-H2 had better survival than HERV-H1 (*p* = 0.0047), and HERV-H3 had better survival than HERV-H1 (*p* = 0.0543). However, no significant differences were observed between HERV-H2 and HERV-H3 (*p* = 0.5767) (Figure 3C).

### 3.5. Retrotranscriptome Quantification Analyses of the HCC Subgroups

Differential expression analysis was performed to better characterize the four HCC subgroups. HERV expression differences were considered significant based on the above-described criteria (adjusted *p*-value < 0.05 and absolute log2-fold change > 1.5). We identified 2365 subgroup-specific HERVs, including 1305 for HERV-H1, 117 for HERV-H2, 876 for HERV-H3, and 67 for HERV-L (Appendix A).

### 3.6. Associations of HCC Subgroups with Survival-Related DE HERVs, Clinical Relevance, Mutations, Copy Number Aberrations, and Structural Variants (SVs)

We further analyzed the associations of the HCC subgroups with the 180 HERVs and found 127 HERVs with statistically significant expression differences among the subgroups and non-cancerous tissues. All 180 HERVs were expressed in cancerous tissues, but five (ERVL-B4-int_1775, PRIMA41-int_0154, HERVL18-int_0054, HERVK22-int_0006, and HERVL-int_0312) were not expressed in non-cancerous tissues (Appendix A).

Next, we ascertained whether the subgroups were associated with clinical parameters. HERV-H2 was more strongly associated with old age than HERV-H1 (*p* = 0.019) (Appendix A), and HERV-H3 was more strongly associated with high AFP than HERV-H2 (*p* = 0.022) (Appendix A).

We further compared the mutations of driver genes among the subgroups (Appendix A). HERV-H1 had more *LRP1B* mutations than HERV-H2 (*p* = 0.048) (Appendix A), HERV-H3 had more *TERT* (*p* = 0.048) and *TP53* (*p* = 0.046) mutations than HERV-L (Appendix A), and the mutation rates of the *TERT* promoter were higher in HERV-H (including H1–H3) than HERV-L (*p* = 0.005) (Appendix A).

We also analyzed the associations of the subgroups with the copy number changes of cancer driver genes. HERV-H2 had more *KDM6A* loss than HERV-H1 (*p* = 0.017), and HERV-L had more *KDM6A* (*p* = 0.028), *TP53* (*p* = 0.049,) and *PER1* (*p* = 0.054) loss than HERV-H1 (Appendix A).

In the analyses of the correlations between SVs and HERV subgroups, SV: 7_63425015_145019722_DUP_1 occurred at a higher rate in HERV-H1 than HERV-H2 (*p* = 0.048), SV: 9_115095253_115095694_TRA_1 occurred at a higher rate in HERV-H1 than HERV-L (*p* = 0.02), SV: 5_11162727_11162728_INS_1 occurred at a higher rate in HERV-H1 (*p* = 0.05) and H3 (*p* = 0.038) than HERV-L, and SV: 5_11162727_11162728_INS_1 occurred at a higher rate in HERV-H (H1–H3) than HERV-L (*p* = 0.033) (Appendix A).

### 3.7. Classification Using HERV Expression in the Cancer Genome Atlas Liver Hepatocellular Carcinoma (TCGA-LIHC)

We used the 180 survival-related DE HERVs and unsupervised approaches to classify the HCCs in TCGA-LIHC and found that the HCCs formed two subgroups, A and B; there was no statistical significance between them (Appendix A). Because our HERV analysis was based on total RNA data, which may not be appropriate for the poly(A)-based data of TCGA, we used the Telescope tool to explore specific differential expression of HERVs in TCGA-LIHC and found 560 DE HERVs between HCCs and non-tumor liver tissues, 103 of which were correlated with survival (Appendix A).

We used these 103 HERVs for classification (Figure 4A) and found that the two subgroups (A and B) had survival differences (*p* = 0.0000014); group B had better survival than group A (Figure 4B). We also found that three HERVs were shared by the TCGA-LIHC and our cohort (Appendix A). We then analyzed the nearby genes of the 103 HERVs in TCGA-LIHC and identified 141 nearby genes. We also found that 11 nearby genes were shared by TCGA-LIHC and our cohort (Appendix A). These analyses demonstrate that using total RNA-seq to explore HERV activation differs from using poly(A) RNA-seq approaches.

### 3.8. Analysis of Host Differential Gene Expression and Molecular Pathways between HERV-H and HERV-L

HERV-H, including HERV-H1–H3, has a higher expression of survival-related 180 HERVs than HERV-L. We collected case data from these two groups to explore the influence of a high amount of survival-related HERVs on the expression of host genes.

There were 2644, 815, and 3220 DE host genes for HERV-H, HERV-L, and both groups, respectively (Figure 5A; Appendix A). Next, we performed gene set enrichment analysis to analyze the pathways of the DE genes for both groups; 50 and 19 pathways were unique for HERV-H and HERV-L, respectively (Appendix A).

Unique pathways for HERV-H included immune and infection, lipid and atherosclerosis, MAPK and NF-kB signaling, cancer and cytokine–cytokine receptor interaction pathways; the mRNA surveillance pathways, nucleocytoplasmic transport, and transcriptional misregulation in cancer pathways were suppressed (Figure 5B).

Unique pathways for HERV-L included metabolism pathways, such as cAMP signaling, amphetamine addiction, calcium signaling, and aldosterone synthesis pathways; necroptosis and ether lipid and glycerophospholipid metabolism pathways were suppressed (Figure 5C).

### 3.9. Analysis of 10 Gene Panels and Nearby Genes of the HCC Subgroups

HERVs are usually inactivated to avoid their influence on host genomic stability via pathways. To evaluate the correlations of the expression levels of related genes among different HCC subgroups, we used MSigDB and recent studies to collect gene panels for HERV activation restriction, KRAB zinc finger proteins (KZFPs), RNA transport, stemness, metabolism, antiviral immunity, human leukocyte antigen (HLA) and antigen processing and presentation (APP), immune checkpoint, inflammasome and inflammatory response, and stimulated three prime antisense retroviral coding sequences (SPARCS). Moreover, nearby HERV activation-related genes may be involved in the development and prognosis of HCC, so we also analyzed the expression levels of 262 nearby genes in different HCC subgroups. The panel of these genes is shown in Appendix A.

The results for all of the HERV-related gene panels are shown in Figure 6; Appendix A. The expression heatmap of the gene panels demonstrates that almost all the genes in the HERV activation restriction, KZFPs, RNA transport, stemness, HLA and APP, and immune checkpoint panels were expressed at lower levels in non-tumor than tumor tissues (Figure 6B; Appendix A). Many genes in the metabolism, antiviral immunity, inflammasome and inflammatory response, SPARCS, and nearby gene panels were also expressed at lower levels in non-tumor than tumor tissues, and few genes in these panels were expressed at higher levels in the non-tumor than tumor tissues (Figure 6B; Appendix A).

We further analyzed differentially expressed genes among the four HCC subgroups and found expression differences (Figure 6C). In the HERV subgroups, the expressions of the gene panels showed no clear differences and only HERV-H3 had highly expressed nearby genes.

For the 96-HERV restriction gene panel, only 13 genes exhibited DE, and 3 (*H3-3A*, *RESF1*, and *IGF2BP1*), 0, 7 (*SMUG*, *KDM1A*, *AGO1*, *FOXM1*, *TET1*, *DNMT3B*, and *IGF2BP3*), and 3 (*GNAS*, *APOBEC3D*, and *PIWIL4*) showed the highest expression for HERV-H1–H3 and L, respectively (Appendix A). Of the 101 *KZFP*s, only 21 showed DE, and all were highly expressed in HERV-H3 (Appendix A). For the 70-RNA transport gene panel, only seven genes had DE among subgroups, and two (*THOC5* and *SRSF12*) and five genes (*NUPs 35*, *93*, and *107*, *THOC3*, and *NDC1*) showed the highest expression in HERV-H1 and HERV-H3, respectively (Appendix A). Of the 126 stemness-related genes, only 29 showed DE, and 11, 1, 16, and 1 had the highest expression for HERV-H1–H3 and L, respectively (Appendix A). Of the 51 metabolic-related genes, 34 showed DE, and 4 (*RHBG*, *SLCO1B3*, *SORD*, and *CYP2C9*), 17, 8, and 5 (*XDH*, *PDE9A*, *CYP11A1*, *CYP2C8*, and *AOX1*) showed the highest expression in HERV-H1–H3 and L, respectively (Appendix A).

Of the 263 antiviral genes, only 27 showed DE, and 10 (PTGES, MAPK13, OLR1, UBA52, MARCKSL1, RNASE2, ZNF503, PHLDA1, IFIT5, and IL12A), 3 (DHX58, ACSL1, and HERC5), 5 (CD276, RPLP0, IKBKE, TRIM5, and SNRNP200), and 9 (GCH1, IL15RA, LY6E, APP, LGALS3BP, IFITM10, CDKN1A, GADD45B, and ETS2) showed the highest expression in HERV-H1–H3 and L, respectively (Appendix A).

For HLA and APP, only one gene (*HLA-E*) showed DE, and had the highest expression in HERV-H1. Of the 22 immune checkpoint genes, only 2 showed DE, and had the highest expression in HERV-H1 (*TNFRSF9*) and H3 (*CD276*), respectively. Of the 54 inflammation and inflammatory genes, only 5 showed DE, and 1 (*C4BPA*) and 4 (*IL32*, *IL4R*, *IL15RA*, and *SIGIRR*) had the highest expression in HERV-H1 and HERV-L, respectively. Of the 15 SPARCs genes, 2 showed DE, and had the highest expression in HERV-H1 (*TRFRSF9*) and L (*IL32*), respectively (Appendix A).

Of the 262 nearby genes, 51 showed DE, and 1 (*RNPS1P1*), 2 (*DTX4* and *LINC02819*), 46, and 2 (*NAV3* and *AC009005.1*) had the highest expression in HERV-H1–H3 and L, respectively (Appendix A).

### 3.10. Associations of the Tumor Microenvironment with Four HCC Subgroups

We used xCell to explore the associations of the HCC subgroups, and 64 immune and stromal cells, with immune, stromal, and microenvironment scores. The numbers of hepatocytes, hematopoietic stem cells (HSC), adipocytes, activated dendritic cells (aDC), preadipocytes, macrophages, monocytes, pericytes, inflammatory (M1) macrophages, reparative (M2) macrophages, epithelial cells, granulocyte-macrophage progenitor (GMP), conventional dendritic cells (cDC), central memory CD4+ T cell (CD4+ Tcm), lymphatic (ly) endothelial cells, class-switched memory B-cells, plasmacytoid dendritic cells (pDC), central memory CD8+ T cell (CD8+ Tcm), megakaryocytes, eosinophils, effector memory CD8+ T cell (CD8+ Tem), keratinocytes, and mesangial cells, and the immune, stromal, and microenvironment scores were higher in non-cancerous than cancerous tissues (Appendix A).

The quantities of hepatocytes, adipocytes, and pro-B cells were higher in the HERV-H than HERV-L group. However, immature dendritic cells (iDC), common lymphoid progenitor (CLP), sebocytes, and neurons were higher in the HERV-L than HERV-H group (Appendix A).

The poorest survival group (HERV-H1) had fewer hepatocytes, adipocytes, and plasma cells, but more mast and mesangial cells, than the better survival group (HERV-H2). The best survival subgroup (HERV-L) had more megakaryocytes than HERV-H3 (Appendix A).

### 3.11. Analysis of the Effects of Splicing-Modulating Drugs on the Expression of HERV

Splicing modulating drugs influence the protein structure of many genes, including histone- and HERV-related genes. Therefore, these drugs may also influence the expression of survival-related HERVs. For confirmation, we reanalyzed the RNA-seq data of a previous study that applied a splicing-modulating compound (BS008) [70]. After BS008 treatment, 20 survival-related HERVs were upregulated, while another 23 survival-related HERVs were downregulated (Figure 7A; Appendix A).

We also analyzed the HERV activation-related gene panels and found changes in their expression patterns after drug therapy (Figure 7B; Appendix A). The expressions of 10 and 36 HERV activation restriction genes were up- and downregulated, respectively. The expression levels of 51 and 6 KZFPs genes were up- and downregulated, respectively. The expression levels of 7 and 20 RNA transport genes were up- and downregulated, respectively. The expression levels of 10 and 38 stemness genes were up- and downregulated, respectively. The expression levels of 5 and 23 metabolism-related genes were up- and downregulated, respectively. The expressions of 67 and 50 antiviral immune genes were up- and downregulated, respectively. The expression levels of 34 and 23 other immune genes were up- and downregulated, respectively, and the expression levels of 39 and 40 nearby genes were up- and downregulated, respectively. A summary of the results is provided in Appendix A.

## 4. Discussion

To explore the impact of HERV expression on cancer, most studies have used poly(A) RNA-based data from TCGA; the use of total RNA has been rare. Solovyov et al. showed that poly(A)-based studies may lose some repeat sequences and are unable to reveal the whole picture of TE alterations in cancer tissues [71]. In this study, we used total RNA and high-depth RNA-seq (100–200 million reads) and the Telescope analytical tool to explore HERV expression in 254 Taiwanese HCCs. We identified 180 DE HERVs correlated with patient survival. We used these 180 HERVs to analyze TCGA-LIHC, but no survival correlations were found. We further used our approach to explore HERV expression in TCGA-LIHC and identified 103 survival-related DE HERVs, 3 of which overlapped with our 180 HERVs. The 103 HERVs were classified into two survival-related subgroups (*p* = 0.0000014). Our study is the first comprehensive, integrated, large-scale analysis to explore the impact of HERV activation on HCCs using total RNA-seq data. In addition, we conducted a comparison between the differentially expressed protein-coding genes identified in our study and the TCGA-LIHC dataset. We revealed that only about 35.2% of differentially expressed protein-coding genes were shared between our study and the TCGA-LIHC dataset, indicating some disparities in gene expression patterns. Therefore, based on our findings, we speculate that the differences observed between total RNA and poly(A) selected RNA not only contribute to variations in the expression of HERVs, but they may also play a role in the differential expression of protein-coding genes. The potential confounders also include the ethnicity of the cohort, sample storage conditions, and the method used for RNA extraction.

Many studies have explored the clinical significance of HERV expression in cancers [9,10,11,22,23,24,72,73,74,75,76,77,78,79]; several studies have used targeted HERVs and immune-related genes, while others have used whole HERV transcriptome approaches and various bioinformatic analysis tools. Recently, an increasing number of studies have begun to use Telescope for whole HERV transcriptome analysis of cancers due to its locus-specific visualization, which can provide results for both locus-specific HERVs and nearby genes [9,74,77,78]. We used similar approaches to analyze 254 Taiwanese HCCs. We found that a low amount of activated HERVs (HERV-L) correlated with better survival than a high amount of activated HERVs (HERV-H–H3). Following the analysis of genomic changes, we found that *LRP1B*, *TERT*, and *TP53* mutations, copy number deletions of *KDM6A*, *TP53*, and *PER1*, and SV 7_63425015_145019722_DUP_1, 9_115095253_115095694_TRA_1, and 5_11162727_11162728_INS_1 were associated with the HERV subgroups. These results suggest that genomic alterations are essential for patient survival and HERV activation. Details on the methods used for the analysis of genomic changes associated with HERV subgroups can be found in our previous study [80].

After understand pathway differences between high- and low-activation HERV subgroups, we systematically analyzed the expression levels of 10 gene panels. Genes with DE in the HERV activation restriction, KZFPs, RNA transport, stemness, HLA and APP, and immune checkpoint panels were over-expressed in the cancerous tissues. The poorer survival HERV-H1 and HERV-H3 subgroups had more highly expressed genes than the better survival subgroups (HERV-L and HERV-H2). These findings contradict the function of HERV activation restriction and KZFP genes, which are expressed in normal cells to suppress HERV expression [18,79,81]. The dysregulation of HERV activation restriction genes has also been found in other cancers, such as the overexpression of *TET1*, *TET3*, and *APOBEC3B* but downregulation of *APOBEC3C*, *3G*, *3D*, *3H*, and *C2* in prostate cancer, and different cancers may involve different activation restriction genes [9,10]. Regarding KZFPs, many of the highly expressed genes in the highest activation HERV subgroup (HERV-H3) have yet to be reported. These results suggest that the dysregulation of genes involved in HERV activation restriction and the HERV sequence binding protein play important roles in HERV activation. The activation of HERVs may result from the activation of oncogenic proteins (such as *TERT*) or downregulation of TSGs (such as *TP53*, *KDM6A*, and *PER1*), and results in KAP1 (TRIM28)-mediated activation of HERV promoters, which increases HERV expression [55,82,83,84,85,86].

The relationship between HERV activation and RNA transport has yet to be explored. Our high HERV activation group (HERV-H3) and poorer survival group (HERV-H1) had more over-expressed nucleocytoplasmic transport genes than the lower HERV activation groups (HERV-L). These results suggest that RNA transport is essential for HERV activation and may be involved in cancer cell survival. Moreover, increased efficiency of nuclear transport of HERV-related RNAs may result in survival benefits for cancer cells. Whether HERV RNA accumulation in the nucleus is toxic for HCC cells like nucleotide-related repeat disorders needs further study [87].

The clinical correlations among HERV activation, host immune-related genes, and therapeutic effects have been heavily studied. This study used panels of 263 antiviral immunity-related genes, 45 HLA and APP-related genes, 22 immune checkpoint-related genes, 54 inflammasome and inflammatory response-related genes, and 15 SPARCS-related genes, and analyzed their correlations with HERV activation and clinical significance. The high HERV activation groups (HERV-H1–H3) had higher expression of immune-suppressing and negative regulator genes, such as *CD276*, *TRIM5*, *OLR1*, and *UBA52*, while higher expression of immunity activators (such as *GADD45B*, *CDKN1A*, *IL12A*, *IL15RA*, *IFIT5*, *IFITM10*, and *LY6E*) was seen for the low HERV activation group (HERV-L). Preexisting host immunity may play a more important role in patient survival, as indicated by Au et al. [24].

The associations between cancer stemness and patient survival are well known, and HERV-K plays a role in promoting and maintaining stem cells in cancer [88]. We systematically analyzed the correlations among 126 stemness-related genes, survival-related HERV activation, and patient survival. Most of the stemness-related genes were over-expressed in cancerous tissues relative to non-cancerous tissues, and only four genes (*PROM1*, *EPCAM*, *HAS2*, and *KRT19*) had higher expression in non-cancerous tissues. There were more over-expressed stemness-related genes in the HERV activation groups (HERV-H1–H3) than in the low HERV activation group (HERV-L; only *ANPEP* showed over-expression). We also analyzed the influence of 51 metabolism-related genes on HERV activation and patient survival and found higher expression of more metabolism-related genes in HERV-H2 and HERV-L than in HERV-H1. From these results, co-analysis of both stemness- and metabolism-related genes may provide a marker for predicting patient survival using HERV activation.

The nearby genes of HERVs have been shown to play important roles in cancer development and other diseases [13]. The reactivation of HERVs leads to the generation of viral transcripts and viral proteins, which can have a significant impact on host gene expression. HERV LTRs, which contain regulatory elements, can act as alternative promoters or enhancers for nearby host genes. When HERVs are reactivated, their LTRs can drive the expression of neighboring host genes, leading to changes in gene expression patterns. This can either result in the upregulation of certain host genes or interfere with the regulation of nearby genes [89]. Seven genes were expressed in non-cancerous tissues only, including *LINC01612* (TSG), *FAM3D* (inflammation), *HCN1*, and *CNBD1* (mutation in cancer) and three other genes. These genes may play a TSG-like role in Taiwanese HCC, and *LINC01612* and *HCN1* were correlated with patient survival. Most nearby genes were over-expressed in cancerous tissues; 64 were upregulated, and 12 were downregulated, in cancerous tissues, including 30 non-coding RNAs, 40 protein coding genes and 6 pseudogenes. Most of these genes influence patient survival via their effects on proliferation (*KCNH8*, *LUCAT1*, and *DLEU1*), metastasis (*MAOA, LINC01612*, and *LINC00578*), invasion (*LINC02163* and *CENPI*), progression (*TRIM24*, *ST8SIA4, KPNA2*, and *CSMD2*), drug resistance (*UGT8*, *SLC22A3*, and *CYSLTR1*), and immunity and immune cells (*DTX4, CHSY3*, and *CCRL2*), and nearby genes are involved in several cancer processes (Appendix A). Twenty-four nearby genes were also associated with patient survival, and HERV-H3 was associated with more activation of nearby genes than HERV-H1, HERV-H2, and HERV-L. These results show that the activation of related nearby HERV-related genes also plays an important role in the development of HCC and patient prognosis.

The tumor microenvironment plays important roles in therapeutic effects and patient survival, particularly immune cell content [90]. Immune phenotyping of Taiwanese HCCs by gene expression analyses of immune and stromal cell markers revealed that non-cancerous tissues have more diverse immune and stromal cells, and higher immune, stromal, and microenvironment scores, than cancerous tissues. Many immune and stromal cells, including iDC, CLP, sebocytes, and neurons, were associated with lower HERV activation and better survival (HERV-L group), but a higher number of hepatocytes, adipocytes, and pro-B cells was associated with poor survival (HERV-H group). The HERV-H1–H3 subgroups were also associated with immune related cells. The better survival (HERV-H2) subgroup had more diverse immune cells than HERV-H1 and HERV-H3, while HERV-H3 had fewer immune cells than HERV-H1 and HERV-H2. This specific immune group may provide a new target for immune therapy.

In this study, we found several types of HERV activations with different transcriptomic changes resulting in survival changes. We suggest that activation or reactivation of HERVs to produce neoantigens, via a demethylation drug combined with immune therapy, may not achieve good survival for all cancer patients. Using a combination of drugs with more comprehensive effects to affect the expression of HERVs and their regulatory genes, and nearby-, stemness-, metabolic-, and antiviral immunity-related genes, may be a more effective way to treat of cancer. We used the splicing-modulating drug BS008 to treat the hepatocellular cell line Huh 7. BS008 splicing regulation involved multiple splicing factors and was accompanied by alterations in the phosphorylation state of serine/arginine-rich proteins (SR proteins) [70]. BS008 can modulate HERV activation, as well as regulatory and nearby genes, and stemness-, metabolism-, and antiviral immunity-related genes. We suggest that BS008 may be essential for HERV-based cancer therapy. However, the lack of patients receiving immunotherapy in our study cohorts restricts our ability to confirm this.

## 5. Conclusions

In conclusion, we identified survival-related DE HERVs in HCC and used these HERVs for molecular classification. Furthermore, our findings revealed significant genomic changes, molecular pathways, 10 gene panels, nearby genes, and alterations in the tumor microenvironment within the different subclasses of HCC. Comprehensive and integrated approaches to evaluate HERV activations in HCC will provide new companion diagnostics and therapeutic strategies, similar to the approaches reported in the study conducted by Steiner et al. [9].

## Figures and Tables

**Figure 1 cancers-15-03664-f001:**
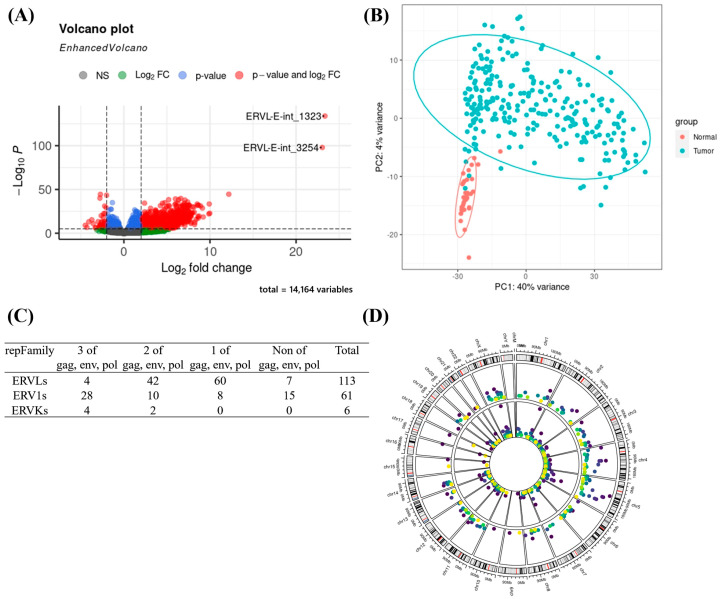
Results of retrotranscriptome analysis. (**A**) Volcano plot of the DE of HERVs in 254 HCCs compared with 34 nearby non-cancerous tissues. (**B**) PCA analysis using the top 500 DE HERVs. (**C**) Sub-families and possible coding protein components of 180 survival-related DE HERVs. (**D**) The circos plot shows the chromosome locations and expression levels of 180 survival-related DE HERVs (outer circle) and 262 nearby genes (inner circle). −log10(P) is plotted. Darker colors corresponding to greater significance.

**Figure 2 cancers-15-03664-f002:**
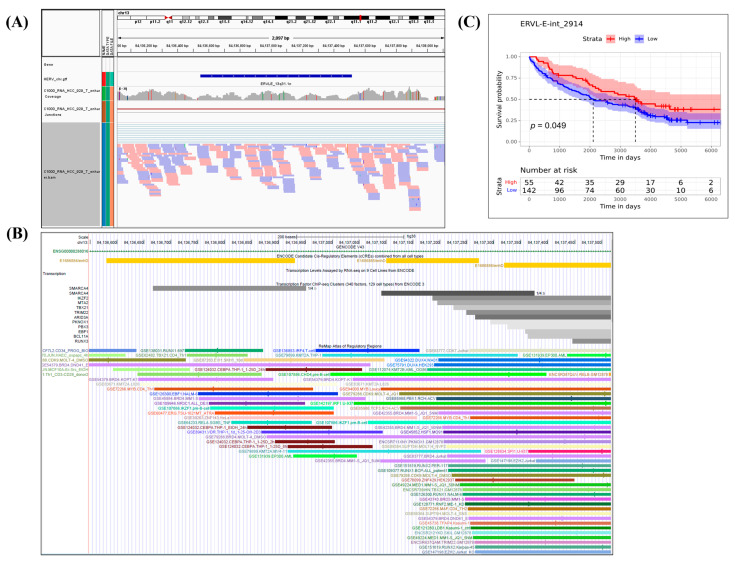
Representative example of a HERV. (**A**) IGV style plot showing the expression of the enhancer ERVL-E-int_2914 in one tumor sample. (**B**) ChIP-seq data confirms the role of the enhancer. (**C**) A KM plot is provided, indicating a potential correlation between this enhancer and patient survival.

**Figure 3 cancers-15-03664-f003:**
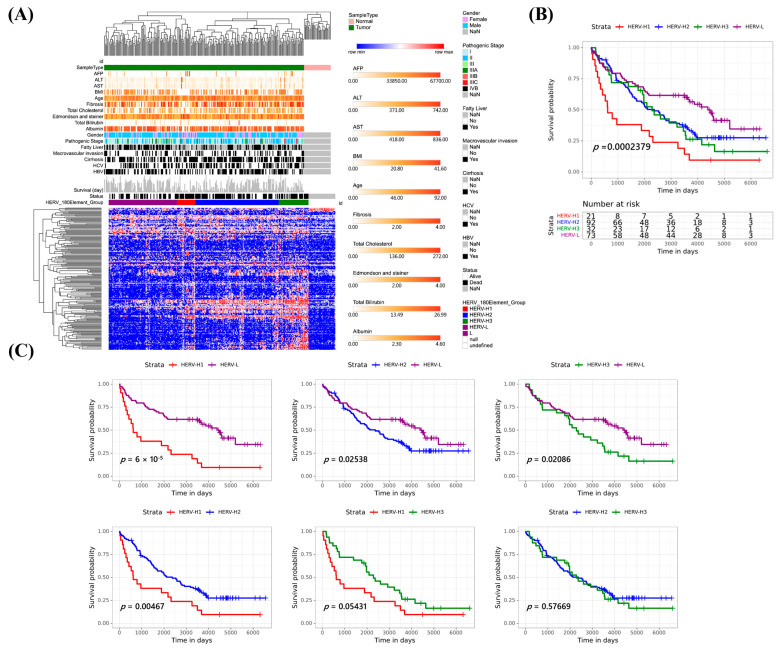
Classification of 254 HCCs using the 180 HERVs transcriptome. (**A**) Unsupervised clustering analysis of 180 HERVs in 254 HCCs and 34 nearby non-cancerous tissues. The heatmap shows four distinct subgroups (H1, H2, H3, and L). (**B**,**C**) Overall survival of each subgroup.

**Figure 4 cancers-15-03664-f004:**
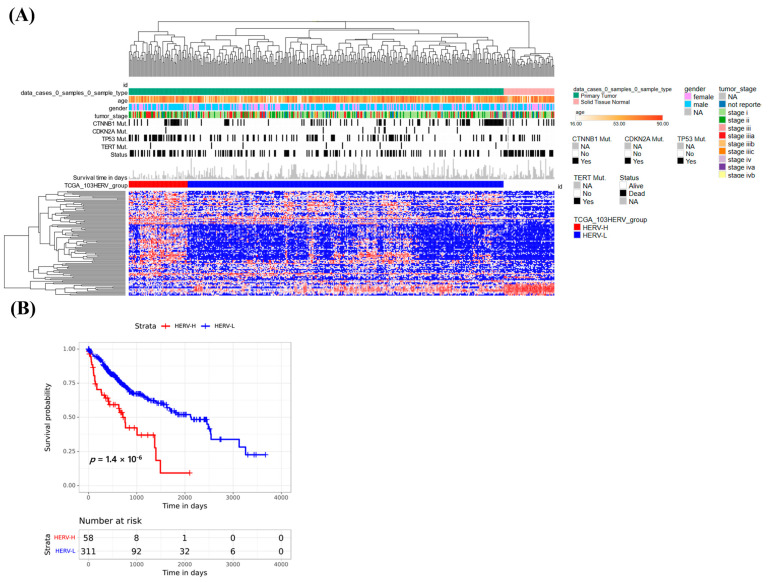
Classification of 365 HCCs from TCGA through transcriptome analysis of 103 HERVs. (**A**) The expression levels of 103 HERVs are indicated by the rows. The heatmap shows the two clusters (**A**,**B**) into which the 103 HERVs were classified, and the top panel shows the clinicopathological and genetic data for each patient (by column) according to the unsupervised clustering. (**B**) Patient survival correlated with the expression patterns of 103 HERVs (*p* = 0.0000014).

**Figure 5 cancers-15-03664-f005:**
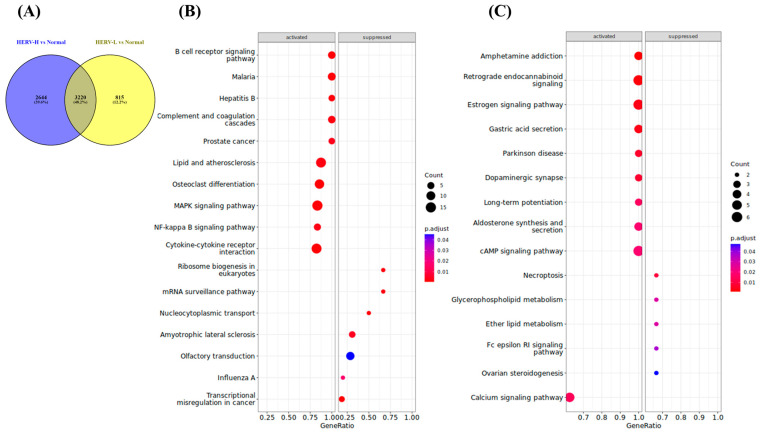
Results of gene set enrichment analysis of subgroup-specific genes. (**A**) Venn diagrams representing the interrelationships of DE genes between HERV-H and HERV-L. (**B**) The unique pathways for HERV-H. (**C**) The unique pathways for HERV-L. Detailed information is listed in Appendix A.

**Figure 6 cancers-15-03664-f006:**
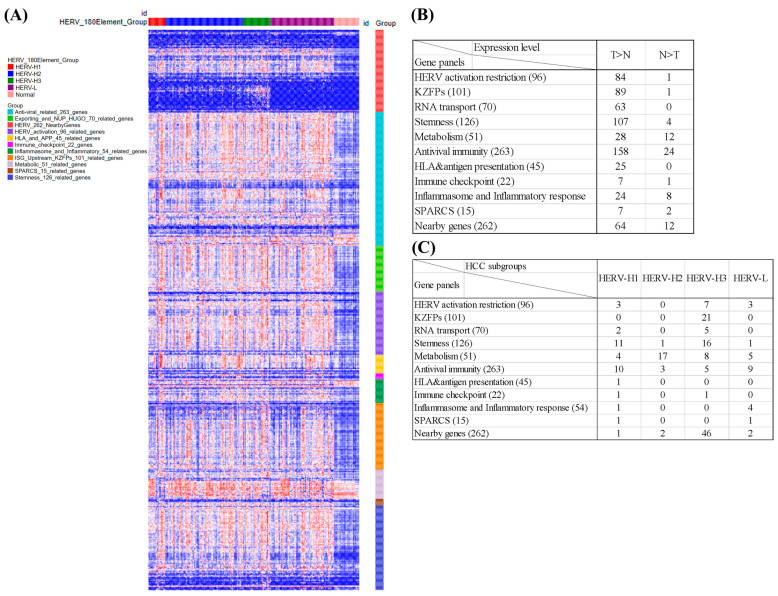
Survival-related DE HERVs among four HCC subgroups and 10 gene panels, and nearby genes, for 254 Taiwanese HCCs and 34 non-cancerous tissues. (**A**) In the heatmap, the expression levels of 180 HERVs subgroups are indicated for each patient in the columns, and gene panel expression data are shown in different colors. (**B**) Summary of count of altered expressions in HERV activation-related genes in HCC and non-tumoral tissues. (**C**) Summary of count of altered expressions in HERV reactivation-related genes in four HCC subgroups.

**Figure 7 cancers-15-03664-f007:**
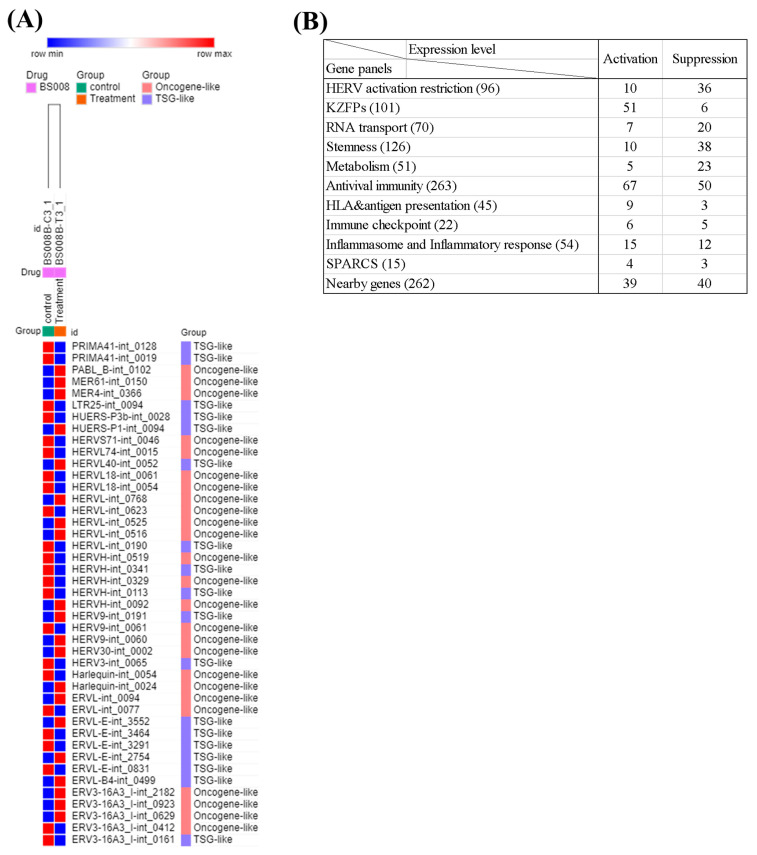
Results of differential gene expression analysis following treatment with BS008. (**A**) Heatmap showing the expression levels of 180 survival-related DE HERVs after drug treatment. Top; column showing the control and drug group data, and right; activated HERVs and their characteristics. (**B**) Summary of count of altered expressions in HERV activation-related genes following drug treatment.

## Data Availability

The WGS and RNA-seq data for this study were submitted to the NCBI Sequence Read Archive under the BioProject PRJNA885992 and PRJNA870935.

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
