# Peer review of "Comprehensive Analysis and Drug Modulation of Human Endogenous Retrovirus in Hepatocellular Carcinomas"

_cancers, 2023, doi:10.3390/cancers15143664_

Round 1

Reviewer 1 Report

This is an interesting manuscript in which the authors described comprehensive analysis of the role of Human endogenous retrovirus (HERV) in hepatocellular carcinomas (HCCs). They performed an extensive collection of analyses utilizing the total RNA sequencing data to analyze the whole HERV alterations in the cancer cells. They observed several types of HERV activations with different transcriptomic changes resulting in survival changes. In general, the paper is convincing and well written. Few more issues could be addressed to further strengthen this yet interesting paper.

1) In line 412, the author mentioned a previous study. Which study should be clarified.

2) The authors stated that BS008 treatment influenced the expression of survival-related HERVs. I suggest the authors perform some experiments to confirm the splicing change based on sequencing.

Author Response

Reviewer #1:

1) In line 412, the author mentioned a previous study. Which study should be clarified.

Response: We have added the reference: “For confirmation, we reanalyzed the RNA-seq data of a previous study that applied a splicing-modulating compound (BS008) [70].”

2) The authors stated that BS008 treatment influenced the expression of survival-related HERVs. I suggest the authors perform some experiments to confirm the splicing change based on sequencing.

Response: Thank you for your comment. The focus of our study is on HERVs; therefore, we did not conduct any splicing change experiments. The splicing changes were carried out in our previous study.

Reviewer 2 Report

very well done paper, however difficult to follow methods and results due to high volume and extent of ninformation presented

Author Response

Reviewer #2:

very well done paper, however difficult to follow methods and results due to high volume and extent of ninformation presented

Response: Thank you for your comment. We have included our sequencing data and provided a description of the methodology in the paper. Please feel free to reach out if you have any questions or if there are any areas that require further clarification.

Reviewer 3 Report

The authors have performed an analysis of HERV expression in 254 HCC patients. The focus of the study is to determine the relevance of HERV expression in HCC in terms of its ability to predict prognosis and define HCC subtypes and their associations with alteration in different biological pathways. While the analysis is reasonably comprehensive, most of the analysis does not go into great depth, meaning that the findings are generally correlative and descriptive without providing mechanistic insights. Nevertheless, the study does provide a new dataset of total RNA-seq in over 250 HCC samples + 34 normals, which will present a valuable resource for the research community. I have some suggestions for additional analysis to improve the validity of the findings. Some improvements could also be made in the presentation of the results:

1.       It should state clearly in the abstract that the comparison is between HCC and normal. The number of HCC and normal samples should be provided, and whether these are paired should be stated. In fact, I also could not find this information in the methods and it is only stated in the legend of Figure 1. Are the 34 normals sampled from non-cancerous tissue from some of the 254 patients with HCC?

2.       If the normal samples are paired, a supplementary analysis should also be performed that compares these paired samples only to ensure that there is good overlap between the differences in the pairs and normal versus tumour in general.

3.       Although the study is concentrated on retrotransposons, it would be very informative to include a PCA plot to determine the difference between tumour and normal based on coding genes.

4.       A lot of the results are presented in supplementary tables. It would be helpful to pick one or two examples and plot out the results in some figure panels, which may include: (1) IGV style plot showing the expression of the HERV across some tumour/normal samples, (2) some ChIP-seq data showing that some of the HERV mentioned is really an enhancer. (3) some KM-plots.

5.       A Cox-regression taking into account the clinical parameters should also be performed in addition to the KM plots.

6.       The authors found that there is very little overlap between the prognostic HERVs from their study and those from TCGA. They suggest that it is due to a difference between total and poly(A) RNA. While this could be a reason, the authors can do more to demonstrate this. For example, they should show that for coding genes, there is a better concordance and that coding genes nearby HERVs that are not consistent between their cohort and TCGA are also consistent. The authors can consider including IGV style tracks for some HERVs that are different between their and TCGA cohort and show that nearby genes are unaffected. More discussion should be added to describe other potential confounders, such as the ethnicity of the cohort, sample storage, RNA extraction method, etc.

7.       For the previous study with the splicing-modulator drug, is the baseline HERV expression concordant with those in this study?

8.       While the study is largely correlative, it would benefit from some discussion on mechanisms of how the HERV is perturbing gene expression or if it is primarily a passenger event related to co-epigenetic dysregulation.

Author Response

Reviewer #3:

  1. It should state clearly in the abstract that the comparison is between HCC and normal. The number of HCC and normal samples should be provided, and whether these are paired should be stated. In fact, I also could not find this information in the methods and it is only stated in the legend of Figure 1. Are the 34 normals sampled from non-cancerous tissue from some of the 254 patients with HCC?

Response: We have added the following sentence to the Abstract: “We employed Telescope to identify HERVs and quantify their expression in the total RNA sequencing data obtained from 254 HCC samples, comprising 254 tumor tissues and 34 matched normal tissues.”

The Methods section of the revised manuscript contains the following addition: “Of these patients, thirty-four had matched tumor and normal tissues.”

  1. If the normal samples are paired, a supplementary analysis should also be performed that compares these paired samples only to ensure that there is good overlap between the differences in the pairs and normal versus tumour in general.

Response: We have added the following sentence to the Results: “Additionally, there are 1,752 DE HERVs in 34 paired HCC samples. Among these 1,752 HERVs, 1607 HERVs (~91.7%) are shared with the 3,357 HERVs found in 288 HCC samples (254 tumor samples and 34 non-tumor samples). There is good overlap between the differences in the pairs and normal versus tumor.”

  1. Although the study is concentrated on retrotransposons, it would be very informative to include a PCA plot to determine the difference between tumour and normal based on coding genes.

Response: We have added the following sentence to the Results: “Moreover, we found protein-coding gene also has the similar phenomenon (details can be provided on request).”

  1. A lot of the results are presented in supplementary tables. It would be helpful to pick one or two examples and plot out the results in some figure panels, which may include: (1) IGV style plot showing the expression of the HERV across some tumour/normal samples, (2) some ChIP-seq data showing that some of the HERV mentioned is really an enhancer. (3) some KM-plots.

Response: We have added Figure 2 in the revision: “Representative example of a HERV. (A) IGV style plot showing the expression of the enhancer ERVL-E-int_1532 in one tumor sample. (B) ChIP-seq data confirms the role of the enhancer. (C) A KM plot is provided, indicating a potential correlation between this enhancer and patient survival.”

  1. A Cox-regression taking into account the clinical parameters should also be performed in addition to the KM plots.

Response: We have added the following sentence to the Methods: “Independent prognostic factors were analyzed by the Cox proportional hazards regression model. Variables in the model included gender, age, and stage.”

We have added the following sentence to the Results: “After adjusting for clinical factors, the classification still demonstrated significant survival implications.”

  1. The authors found that there is very little overlap between the prognostic HERVs from their study and those from TCGA. They suggest that it is due to a difference between total and poly(A) RNA. While this could be a reason, the authors can do more to demonstrate this. For example, they should show that for coding genes, there is a better concordance and that coding genes nearby HERVs that are not consistent between their cohort and TCGA are also consistent. The authors can consider including IGV style tracks for some HERVs that are different between their and TCGA cohort and show that nearby genes are unaffected. More discussion should be added to describe other potential confounders, such as the ethnicity of the cohort, sample storage, RNA extraction method, etc.

Response: We have added the following sentence to the Discussion: “In addition, we conducted a comparison between the differentially expressed protein-coding genes identified in our study and the TCGA-LIHC dataset. We revealed that only about 35.2% of differentially expressed protein-coding genes were shared between our study and the TCGA-LIHC dataset, indicating some disparities in gene expression patterns. Therefore, based on our findings, we speculate that the differences observed between total RNA and poly(A) selected RNA not only contribute to variations in the expression of HERVs, but they may also play a role in the differential expression of protein-coding genes. The potential confounders also include the ethnicity of the cohort, sample storage conditions, and the method used for RNA extraction.”

  1. For the previous study with the splicing-modulator drug, is the baseline HERV expression concordant with those in this study?

Response: Yes, the analysis of HERV expression in the previous study with the splicing-modulator drug is concordant with this study.

  1.  While the study is largely correlative, it would benefit from some discussion on mechanisms of how the HERV is perturbing gene expression or if it is primarily a passenger event related to co-epigenetic dysregulation.

Response: We have added the following sentence to the Discussion: “The reactivation of HERVs leads to the generation of viral transcripts and viral proteins, which can have a significant impact on host gene expression. HERV LTRs, which contain regulatory elements, can act as alternative promoters or enhancers for nearby host genes. When HERVs are reactivated, their LTRs can drive the expression of neighboring host genes, leading to changes in gene expression patterns. This can either result in the upregulation of certain host genes or interfere with the regulation of nearby genes [90].”

Reviewer 4 Report

The article presents a comprehensive analysis of HERV alterations in HCCs, utilizing total RNA sequencing data from 254 Taiwanese patients. The authors successfully identified 3,357 locus-specific activations of HERVs that exhibited differential expression, with 180 of them being correlated with patient survival. Additionally, the study revealed the presence of distinct subgroups within HCCs based on HERV expression patterns, shedding light on potential therapeutic strategies. The comments below are to improve already excellent MS.

Comments for authors:

Title:

The title accurately reflects the content of the manuscript and is concise.

Abstract:

1.      The abstract lacks specific details on how the study findings could contribute to companion diagnostics and therapeutic strategies. Providing a more explicit and concrete explanation of these potential contributions would improve the abstract's impact.

2.      A concise summary of the study objectives, methods, and main findings is needed to provide readers with a clear overview. This should include key results and their implications.

3.      The direction and magnitude of the correlations between HERV activation and certain genes and pathways should be provided to enhance the interpretation of the results.

4.      Mentioning the specific bioinformatic tools used for HERV analysis, along with appropriate references or additional information, would promote reproducibility.

Introduction:

5.      The introduction lacks a clear background or context for the study. Briefly explaining the significance of HERV in cancer development and highlighting existing literature gaps that the study aims to address would be beneficial.

6.      A concise summary of the study objectives, methods, and main findings should be included in the introduction to provide readers with a clear overview.

7.      The relevance of the mentioned risk factors in the context of HERV activation and their impact on HCC development needs to be elaborated upon.

8.      The importance of studying HERVs in HCC and their potential implications should be emphasized, providing a clear transition and justification for focusing on HERVs in the study.

9.      Providing more details and specific references to support the statement that reactivating HERVs using demethylation drugs is a new approach for cancer treatment would strengthen the introduction. Explaining the mechanisms by which demethylation drugs induce HERV reactivation, promote neoantigen production, and elicit anti-viral-like immunity is essential.

Methods: Liver samples and clinical data:

10.   Specific information about the selection criteria for liver samples used in the study, such as tumor stage, patient demographics, or other relevant factors, should be provided to enhance transparency.

RNA extraction and RNA sequencing (RNA-seq):

11.   The RNA extraction and sequencing methodology are adequately described, including the use of appropriate kits and instruments.

12.   It would be beneficial to provide additional information about the number of samples subjected to RNA-seq and any implemented quality control measures to ensure the reliability of the sequencing data.

Metatranscriptome analysis:

13.   No details are provided regarding the specific objectives or findings of the metatranscriptome analysis. Briefly explaining how this analysis relates to the overall study goals and the insights it provides would be helpful.

Retrotranscriptome and transcriptome quantification:

14.   The rationale for focusing on HERVs and the specific research questions related to their role in HCC should be explicitly stated, providing a brief background or motivation for studying HERVs in this context.

15.   The main findings or insights obtained from the analysis of the structure of survival-related HERVs, their genomic regions, and nearby genes should be mentioned.

Molecular classification:

16.   Details on the specific methodology or algorithms used for molecular classification should be provided, along with an overview of its significance in the study.

17.   Correlation analysis of HERV subgroups and the expression levels of genes related to HERV restriction, viral immunity, RNA transport, and stemness:

18.   Briefly explaining the rationale for selecting these specific gene panels and how their correlation with HERVs relates to the research objectives of the study would enhance this section.

Tumor microenvironment analysis:

19.   No specific details are provided regarding the purpose or findings of the tumor microenvironment analysis. Briefly mentioning the relevance of this analysis to the study and any significant observations would be beneficial.

Drug treatment of HCC cell line:

20.   The connection between drug treatment of HCC cell lines and the main study objectives or research questions should be clarified. Providing a brief rationale for performing this experiment and its potential implications would improve this section.

Differential expression analysis:

21.   Describing the normalization strategy and providing cutoff criteria for identifying differentially expressed genes would enhance the rigor of the analysis.

Results:

22.   Retrotranscriptome and Transcriptome Quantification Analyses:

23.   Out of the differentially expressed HERVs, 180 were significantly correlated with patient survival.

24.   Expanding on the specific subtypes (ERVLs, ERV1s, ERVKs) and retroviral components (gag, env, pol) represented among the 180 survival-related HERVs would provide additional insights.

Discussion:

25.   Providing a proper context for the relevance of HERV expression in cancer and discussing the existing literature on this topic is essential.

26.   Discussing the advantages and limitations of using total RNA and high-depth RNA-seq methods would provide a comprehensive understanding of the applied techniques.

27.   Providing a clear description of the Telescope analytical tool and its specific functionalities in analyzing HERV expression is necessary.

28.   Details on the methods used and specific findings related to the analysis of genomic changes associated with HERV subgroups should be included.

29.   Comparing the advantages and limitations of total RNA and high-depth RNA-seq with poly(A) RNA-based data would enrich the discussion.

30.   Elaborating on the relationship between HERV activation and RNA transport, supported by evidence and potential mechanisms, would enhance the section.

31.   Discussing the clinical correlations among HERV activation, host immune-related genes, and therapeutic effects with specific details and references is recommended.

32.   Elaborating on the relationship between the tumor microenvironment, HERV activation, and patient survival, along with supporting evidence and potential implications, would strengthen the discussion.

33.   Explaining the mechanism of action and previous studies supporting the use of the splicing-modulating drug BS is necessary.

34.   In the discussion of companion diagnostics and therapeutic strategies, citing relevant studies or examples where similar approaches have been applied in cancer research or related fields would provide a comprehensive view.

Conclusions:

35.   Begin the conclusions section by summarizing the main findings of the study in a clear and concise manner.

36.   Citing specific studies or mechanisms previously reported in the literature when discussing the influence of genomic changes on HERV activation would strengthen the conclusions.

37.   Referring to existing studies or methods used successfully in similar contexts when proposing comprehensive approaches for evaluating HERV activations would enhance the conclusions.

Language and Style:

The manuscript is written in a manner befitting a scientific research study. The findings and their consequences can be better communicated to the target audience when written in clear, succinct, and objective language.

Figures and Tables:

All tables and figures are legible, have enough labels, and contribute to the study. More descriptive captions that provide clear summaries of the facts would be beneficial. In addition, the overall presentation of the tables and figures in the manuscript might be enhanced by maintaining uniformity in formatting and style throughout.

Author Response

Reviewer #4:

Title:

The title accurately reflects the content of the manuscript and is concise.

Response: Thank you for your comment.

Abstract:

  1. The abstract lacks specific details on how the study findings could contribute to companion diagnostics and therapeutic strategies. Providing a more explicit and concrete explanation of these potential contributions would improve the abstract's impact.

Response: The sentence of conclusion in the abstract has been rewritten as follows: “Comprehensive and integrated approaches for evaluating HERV expression and their correlation with specific pathways have the potential to provide new companion diagnostics and therapeutic strategies for HCC.”

  1. A concise summary of the study objectives, methods, and main findings is needed to provide readers with a clear overview. This should include key results and their implications.

Response: The Simple Summary has been rewritten as follows: “Human endogenous retrovirus (HERV) plays important roles in the development of cancer, and most studies using data of The Cancer Genome Atlas to analyze the whole HERV alterations in the cancer cells. For HCC, most studies have focused on LINE-1 and specific HERV to explore their importance. In this study, we used our total RNA sequencing data of 254 Taiwanese HCCs and many bioinformatic tools to analyze HERV alterations, and then explored the correlations between HERV activation and pathways and certain gene panels. Unique pathways for higher expression of survival-related HERVs included immune and infection, lipid and atherosclerosis, MAPK and NF-kB signaling, and cytokine-cytokine receptor interaction pathways were activated; the mRNA surveillance pathway, nucleocytoplasmic transport, ribosome biogenesis, and transcriptional misregulation in cancer pathways were suppressed. We found that many overexpressed HERV-related nearby genes were correlated with high HERV activation and poor survival. The implementation of comprehensive and integrated approaches to assess HERV expression and their association with specific pathways is poised to offer novel companion diagnostics and therapeutic strategies for HCC.”

  1. The direction and magnitude of the correlations between HERV activation and certain genes and pathways should be provided to enhance the interpretation of the results.

Response: The Abstract section of the revised manuscript contains the following addition: “The differential expression of host genes in high expression of these 180 HERVs primarily involved the activation of pathways related to immune and infection, lipid and atherosclerosis, MAPK and NF-kB signaling, and cytokine-cytokine receptor interaction. Conversely, there was a suppression of pathways associated with RNA processing, including nucleocytoplasmic transport, surveillance and ribosome biogenesis, and transcriptional misregulation in cancer pathways.”

  1. Mentioning the specific bioinformatic tools used for HERV analysis, along with appropriate references or additional information, would promote reproducibility.

Response: The sentence has been rewritten as follows: “We employed Telescope to identify HERVs and quantify their expression in the total RNA sequencing data obtained from 254 HCC samples, comprising 254 tumor tissues and 34 matched normal tissues.

Introduction:

  1. The introduction lacks a clear background or context for the study. Briefly explaining the significance of HERV in cancer development and highlighting existing literature gaps that the study aims to address would be beneficial.

Response: The Introduction section of the revised manuscript contains the following addition: “Reactivation of HERV is found in many cancers and can influence tumor genome stability [9-12]. HERVs can serve as alternative promoters or enhancers for nearby genes in malignant cells, inducing both tumor suppressor gene (TSG) downregulation and oncogene upregulation, and cryptic transcription start sites within HERVs can be employed to produce aberrant protein-coding mRNAs [13-15]. These alterations result in cancer development, progression, metastasis, immune alterations, and chemoresistance [9-12,14-18]. Retrotransposons are correlated with the development of HCC, but most studies have focused on LINE 1, and rarely on whole HERVs [19,20].”

  1. A concise summary of the study objectives, methods, and main findings should be included in the introduction to provide readers with a clear overview.

Response: The Introduction section of the revised manuscript contains the following addition: “In this study, we employed Telescope to conduct a locus-specific characterization of survival-related differentially expressed (DE) HERVs in HCC. These survival-related DE HERVs were used to molecular classification. Subsequently, we investigated the correlation between HERV subgroups and the expression levels of genes associated with HERV restriction, viral immunity, RNA transport, stemness, nearby genes and the tumor microenvironment. Finally, we analyzed the impact of splicing-modulating drugs on the expression of HERVs.”

  1. The relevance of the mentioned risk factors in the context of HERV activation and their impact on HCC development needs to be elaborated upon.

Response: We have added the following sentence to the Introduction: “Reactivation of HERVs in cancer cells may result in viral mimicry state, generation of highly tumor-specific antigens and expression of long terminal repeats (LTR)-activated transcripts and its implications for cancer immunotherapy [21].”

  1. The importance of studying HERVs in HCC and their potential implications should be emphasized, providing a clear transition and justification for focusing on HERVs in the study.

Response: The Introduction section of the revised manuscript contains the following addition: “Retrotransposons are correlated with the development of HCC, but most studies have focused on LINE 1, and rarely on whole HERVs [19,20].”

  1. Providing more details and specific references to support the statement that reactivating HERVs using demethylation drugs is a new approach for cancer treatment would strengthen the introduction. Explaining the mechanisms by which demethylation drugs induce HERV reactivation, promote neoantigen production, and elicit anti-viral-like immunity is essential.

Response: The Introduction section of the revised manuscript contains the following addition: “For example, DNA-demethylation agents have shown clinical anti-tumor efficiency by inducing transcription of endogenous dsRNAs that activate the viral recognition and interferon response pathway in colorectal cancer-initiating cells [26].”

Methods: Liver samples and clinical data:

  1. Specific information about the selection criteria for liver samples used in the study, such as tumor stage, patient demographics, or other relevant factors, should be provided to enhance transparency.

Response: The Results section of the revised manuscript contains the following addition: “The demographic data of the 254 HCC patients are shown in Table S1.”

RNA extraction and RNA sequencing (RNA-seq):

  1. The RNA extraction and sequencing methodology are adequately described, including the use of appropriate kits and instruments.

Response: Total RNA was extracted from tissue samples using the NucleoSpin® RNA Kit (Ma-cherey–Nagel, Duren, Germany), following the manufacturer’s instructions. The quality, quantity, and integrity of the total RNA were evaluated using the NanoDrop 1000 spec-trophotometer and Bioanalyzer 2100 (Agilent Technologies, Santa Clara, CA, USA). RNA-seq was performed as described previously [27]. Briefly, samples with an RNA in-tegrity number > 6.0 were used for RNA-seq. A barcoded library was generated using a Total RNA Library Preparation Kit (Illumina, San Diego, CA, USA). The libraries were se-quenced on a NovaSeq 6000 instrument (Illumina) using 2 × 151-bp paired-end sequenc-ing flow cells following the manufacturer’s instructions. A more detailed description is now given in the revision (Methods: RNA extraction and RNA sequencing section).

  1. It would be beneficial to provide additional information about the number of samples subjected to RNA-seq and any implemented quality control measures to ensure the reliability of the sequencing data.

Response: We have added the following sentence to the Methods: “Samples with an RNA integrity number > 6.0 were used for RNA-seq.”

Metatranscriptome analysis:

  1. No details are provided regarding the specific objectives or findings of the metatranscriptome analysis. Briefly explaining how this analysis relates to the overall study goals and the insights it provides would be helpful.

Response: The Methods section of the revised manuscript contains the following addition: “To characterize the microbiome composition in tumors, we used Kraken2 (v2.1.1) for the analysis of metatranscriptomic data.”

The Results section of the revised manuscript contains the following addition: “The metatranscriptome results revealed 109 viral RNAs in 254 HCCs, including 102 HBVs and 7 HCVs (Table S2).”

Retrotranscriptome and transcriptome quantification:

  1. The rationale for focusing on HERVs and the specific research questions related to their role in HCC should be explicitly stated, providing a brief background or motivation for studying HERVs in this context.

Response: The Introduction section of the revised manuscript contains the following addition: “Retrotransposons are correlated with the development of HCC, but most studies have focused on LINE 1, and rarely on whole HERVs [19,20].”

  1. The main findings or insights obtained from the analysis of the structure of survival-related HERVs, their genomic regions, and nearby genes should be mentioned.

Response: The Results section of the revised manuscript contains the following addition: “We used HERV meta-annotations provided by Telescope to examine the structural and functional properties of DE elements and nearby or intersected genes. Of the 180 survival-related DE HERVs, 46, 116, and 18 were in intronic, intergenic and exonic regions, respectively; 12 of these 180 HERVs contained protein coding transcripts. Two HERVs in intronic regions, and one in an intergenic region, were enhancers (Table S5).”

Molecular classification:

  1. Details on the specific methodology or algorithms used for molecular classification should be provided, along with an overview of its significance in the study.

Response: The Methods section of the revised manuscript contains the following addition: “For molecular classification, an unsupervised approach was used. Each signature HERV was significantly up- or downregulated in one subclass relative to the other subclasses. The TPM value was used as the distance metric (one minus spearman rank correlation), and the average linkage method was applied. Signature HERVs and subjects within each class were hierarchically clustered.”

  1. Correlation analysis of HERV subgroups and the expression levels of genes related to HERV restriction, viral immunity, RNA transport, and stemness:

Response: The results of the correlation analysis of HERV subgroups and the expression levels of genes related to HERV restriction, viral immunity, RNA transport, and stemness have been presented in Result section 3.9.

  1. Briefly explaining the rationale for selecting these specific gene panels and how their correlation with HERVs relates to the research objectives of the study would enhance this section.

Response: The Results section of the revised manuscript contains the following addition: “HERVs are usually inactivated to avoid their influence on host genomic stability via pathways. To evaluate the correlations of the expression levels of related genes among different HCC subgroups, we used MSigDB and recent studies to collect gene panels for HERV activation restriction, KRAB zinc finger proteins (KZFPs), RNA transport, stemness, metabolism, antiviral immunity, human leukocyte antigen (HLA) and antigen processing and presentation (APP), immune checkpoint, inflammasome and inflammatory response, and stimulated 3 prime antisense retroviral coding sequences (SPARCS). Moreover, nearby HERV activation-related genes may be involved in the development and prognosis of HCC, so we also analyzed the expression levels of 262 nearby genes in different HCC subgroups. The panel of these genes is shown in Table S20.

The results of the correlation analysis of HERV subgroups and the expression levels of genes related to HERV restriction, viral immunity, RNA transport, and stemness have been presented in Result section 3.9.

Tumor microenvironment analysis:

  1. No specific details are provided regarding the purpose or findings of the tumor microenvironment analysis. Briefly mentioning the relevance of this analysis to the study and any significant observations would be beneficial.

Response: The Methods section of the revised manuscript contains the following addition: “For cell enrichment analysis of various immune and stromal cells in tumors, xCell was used.”

The results of the tumor microenvironment analysis have been presented in Result section 3.10.

Drug treatment of HCC cell line:

  1. The connection between drug treatment of HCC cell lines and the main study objectives or research questions should be clarified. Providing a brief rationale for performing this experiment and its potential implications would improve this section.

Response: The Methods section of the revised manuscript contains the following addition: “In order to characterize the potential of amiloride derivative 3,5-diamino-6-chloro-N-(N-(2,6-dichlorobenzoyl)carbamimidoyl)pyrazine-2-carboxide (BS008) for HERV-based cancer therapy, we treated the Huh-7 cells (HCC-related cell line) with BS008.”

Differential expression analysis:

  1. Describing the normalization strategy and providing cutoff criteria for identifying differentially expressed genes would enhance the rigor of the analysis.

Response: The Methods section of the revised manuscript contains the following addition: “The transcripts per million (TPM) was utilized as the normalization method for RNA-seq data.” and “The absolute log2 Fold Change >1.5 and adjusted p value <0.05 were defined as differentially expressed genes.”

Results:

  1. Retrotranscriptome and Transcriptome Quantification Analyses:
  2. Out of the differentially expressed HERVs, 180 were significantly correlated with patient survival.
  3. Expanding on the specific subtypes (ERVLs, ERV1s, ERVKs) and retroviral components (gag, env, pol) represented among the 180 survival-related HERVs would provide additional insights.

Response: The specific subtypes (ERVLs, ERV1s, ERVKs) and retroviral components (gag, env, pol) represented among 180 survival-related HERVs has been added in Table S4.

Discussion:

  1. Providing a proper context for the relevance of HERV expression in cancer and discussing the existing literature on this topic is essential.

Response: The Discussion section of the revised manuscript contains the following addition: “Many studies have explored the clinical significance of HERV expression in cancers [9-11,22-24,72-79]; several studies have used targeted HERVs and immune-related genes, while others have used whole HERV transcriptome approaches and various bioinformatic analysis tools. Recently, an increasing number of studies have begun to use Telescope for whole HERV transcriptome analysis of cancers due to its locus-specific visualization, which can provide results for both locus-specific HERVs and nearby genes [9,74,77,78].”

  1. Discussing the advantages and limitations of using total RNA and high-depth RNA-seq methods would provide a comprehensive understanding of the applied techniques.

Response: The Discussion section of the revised manuscript contains the following addition: “To explore the impact of HERV expression on cancer, most studies have used poly(A) RNA-based data from TCGA; the use of total RNA has been rare. Solovyov et al. showed that poly(A)-based studies may lose some repeat sequences and are unable to reveal the whole picture of TE alterations in cancer tissues [70]. In this study, we used total RNA and high-depth RNA-seq (100–200 million reads) and the Telescope analytical tool to explore HERV expression in 254 Taiwanese HCCs. We identified 180 DE HERVs correlated with patient survival. We used these 180 HERVs to analyze TCGA-LIHC, but no survival correlations were found. We further used our approach to explore HERV expression in TCGA-LIHC and identified 103 survival-related DE HERVs, 3 of which overlapped with our 180 HERVs. The 103 HERVs were classified into two survival-related subgroups (p = 1.4e-06). Our study is the first comprehensive, integrated, large-scale analysis to explore the impact of HERV activation on HCCs using total RNA-seq data.”

  1. Providing a clear description of the Telescope analytical tool and its specific functionalities in analyzing HERV expression is necessary.

Response: The Discussion section of the revised manuscript contains the following addition: “Recently, an increasing number of studies have begun to use Telescope for whole HERV transcriptome analysis of cancers due to its locus-specific visualization, which can provide results for both locus-specific HERVs and nearby genes [9,74,77,78].”

  1. Details on the methods used and specific findings related to the analysis of genomic changes associated with HERV subgroups should be included.

Response: The Discussion section of the revised manuscript contains the following addition: “Details on the methods used to the analysis of genomic changes associated with HERV subgroups according our previous study [80].”

Details on the specific findings related to the analysis of genomic changes associated with HERV subgroups is now given in the revision (Results section 3.6).

  1. Comparing the advantages and limitations of total RNA and high-depth RNA-seq with poly(A) RNA-based data would enrich the discussion.

Response: The Discussion section of the revised manuscript contains the following addition: “To explore the impact of HERV expression on cancer, most studies have used poly(A) RNA-based data from TCGA; the use of total RNA has been rare. Solovyov et al. showed that poly(A)-based studies may lose some repeat sequences and are unable to reveal the whole picture of TE alterations in cancer tissues [70]. In this study, we used total RNA and high-depth RNA-seq (100–200 million reads) and the Telescope analytical tool to explore HERV expression in 254 Taiwanese HCCs. We identified 180 DE HERVs correlated with patient survival. We used these 180 HERVs to analyze TCGA-LIHC, but no survival correlations were found. We further used our approach to explore HERV expression in TCGA-LIHC and identified 103 survival-related DE HERVs, 3 of which overlapped with our 180 HERVs. The 103 HERVs were classified into two survival-related subgroups (p = 1.4e-06). Our study is the first comprehensive, integrated, large-scale analysis to explore the impact of HERV activation on HCCs using total RNA-seq data.”

  1. Elaborating on the relationship between HERV activation and RNA transport, supported by evidence and potential mechanisms, would enhance the section.

Response: The Discussion section of the revised manuscript contains the following addition: “The relationship between HERV activation and RNA transport have yet to be explored. Our high HERV activation group (HERV-H3) and poorer survival group (HERV-H1) had more over-expressed nucleocytoplasmic transport genes than the lower HERV activation groups (HERV-L). These results suggest that RNA transport is essential for HERV activation and may be involved in cancer cell survival. Moreover, increased efficiency of nuclear transport of HERV-related RNAs may result in survival benefits for cancer cells. Whether HERV RNA accumulation in the nucleus is toxic for HCC cells like nucleotide-related repeat disorders needs further study [88].”

  1. Discussing the clinical correlations among HERV activation, host immune-related genes, and therapeutic effects with specific details and references is recommended.

Response: The Discussion section of the revised manuscript contains the following addition: “The clinical correlations among HERV activation, host immune-related genes, and therapeutic effects have been heavily studied. This study used panels of 263 antiviral immunity-related genes, 45 HLA and APP-related genes, 22 immune checkpoint-related genes, 54 inflammasome and inflammatory response-related genes, and 15 SPARCS-related genes, and analyzed their correlations with HERV activation and clinical significance. The high HERV activation groups (HERV-H1–H3) had higher expression of immune-suppressing and negative regulator genes, such as CD276, TRIM5, OLR1, and UBA52, while higher expression of immunity activators (such as GADD45B, CDKN1A, IL12A, IL15RA, IFIT5, IFITM10, and LY6E) was see for the low HERV activation group (HERV-L). Preexisting host immunity may play a more important role in patient survival, as indicated by Au et al. [24].”

  1. Elaborating on the relationship between the tumor microenvironment, HERV activation, and patient survival, along with supporting evidence and potential implications, would strengthen the discussion.

Response: The Discussion section of the revised manuscript contains the following addition: “The tumor microenvironment plays important roles in therapeutic effects and patient survival, particularly immune cell content [90]. Immune phenotyping of Taiwanese HCCs by gene expression analyses of immune and stromal cell markers revealed that non-cancerous tissues have more diverse immune and stromal cells, and higher immune, stromal, and microenvironment scores, than cancerous tissues. Many immune and stromal cells, including iDC, CLP, sebocytes, and neurons, were associated with lower HERV activation and better survival (HERV-L group), but a higher number of hepatocytes, adipocytes, and pro-B cells was associated with poor survival (HERV-H group). The HERV-H1–H3 subgroups were also associated with immune related cells. The better sur-vival (HERV-H2) subgroup had more diverse immune cells than HERV-H1 and HERV-H3, while HERV-H3 had fewer immune cells than HERV-H1 and HERV-H2. This specific immune group may provide a new target for immune therapy.”

  1. Explaining the mechanism of action and previous studies supporting the use of the splicing-modulating drug BS is necessary.

Response: The Discussion section of the revised manuscript contains the following addition: “BS008 splicing regulation involved multiple splicing factors and was accompanied by al-terations in the phosphorylation state of serine/arginine-rich proteins (SR proteins) [70].”

  1. In the discussion of companion diagnostics and therapeutic strategies, citing relevant studies or examples where similar approaches have been applied in cancer research or related fields would provide a comprehensive view.

Response: Comprehensive and integrated approaches to evaluate HERV activations in HCC will provide a new companion diagnostics and therapeutic strategies, similar to the approaches reported in the study conducted by Steiner et al. [9]. This sentence has added into the revision.

Conclusions:

  1. Begin the conclusions section by summarizing the main findings of the study in a clear and concise manner.

Response: The Conclusions section of the revised manuscript contains the following addition: “In conclusion, we identified survival-related DE HERVs in HCC and used these HERVs for molecular classification. Furthermore, our findings revealed significant genomic changes, molecular pathways, 10 gene panels, nearby genes, and alterations in the tumor microenvironment within the different subclasses of HCC.”

  1. Citing specific studies or mechanisms previously reported in the literature when discussing the influence of genomic changes on HERV activation would strengthen the conclusions.

Response: The Conclusions section of the revised manuscript contains the following addition: “Comprehensive and integrated approaches to evaluate HERV activations in HCC will provide a new companion diagnostics and therapeutic strategies, similar to the approaches reported in the study conducted by Steiner et al. [9].”

  1. Referring to existing studies or methods used successfully in similar contexts when proposing comprehensive approaches for evaluating HERV activations would enhance the conclusions.

Response: The Conclusions section of the revised manuscript contains the following addition: Comprehensive and integrated approaches to evaluate HERV activations in HCC will provide a new companion diagnostics and therapeutic strategies, similar to the approaches reported in the study conducted by Steiner et al. [9].”

Language and Style:

The manuscript is written in a manner befitting a scientific research study. The findings and their consequences can be better communicated to the target audience when written in clear, succinct, and objective language.

Response: Thank you for your comment.

Figures and Tables:

All tables and figures are legible, have enough labels, and contribute to the study. More descriptive captions that provide clear summaries of the facts would be beneficial. In addition, the overall presentation of the tables and figures in the manuscript might be enhanced by maintaining uniformity in formatting and style throughout.

Response: Thank you for your comment.

Round 2

Reviewer 1 Report

In general, the paper is convincing and well written. Data sustains the hypothesis of the authors. I believe that it is an interesting paper, suitable for publication in Cancers.